# T cell toxicity induced by tigecycline binding to the mitochondrial ribosome

Qiuya Shao[1,2,8], Anas Khawaja [2,8], Minh Duc Nguyen [2,3,8], Vivek Singh [2], Jingdian Zhang[2], Yong Liu [4], Joel Nordin [4], Monika Adori[5], C. Axel Innis [6], Xaquin Castro Dopico [5,7] ✉ & Joanna Rorbach [2] ✉

Tetracyclines are essential bacterial protein synthesis inhibitors under continual development to combat antibiotic resistance yet suffer from unwanted side effects. Mitoribosomes - responsible for generating oxidative phosphorylation (OXPHOS) subunits - share structural similarities with bacterial machinery and may suffer from cross-reactivity. Since lymphocytes rely upon OXPHOS upregulation to establish immunity, we set out to assess the impact of ribosome-targeting antibiotics on human T cells. We find tigecycline, a third-generation tetracycline, to be the most cytotoxic compound tested. In vitro, 5–10 µM tigecycline inhibits mitochondrial but not cytosolic translation, mitochondrial complex I, III and IV expression, and curtails the activation and expansion of unique T cell subsets. By cryo-EM, we find tigecycline to occupy three sites on T cell mitoribosomes. In addition to the conserved A-site found in bacteria, tigecycline also attaches to the peptidyl transferase center of the large subunit. Furthermore, a third, distinct binding site on the large subunit, aligns with helices analogous to those in bacteria, albeit lacking methylation in humans. The data provide a mechanism to explain part of the anti-inflammatory effects of these drugs and inform antibiotic design.

Anti-microbial resistance is a major One Health problem that has led to an intensive search for new antibacterials, as well as the modification of existing entities to broaden their spectrum of activity. Many antibiotics on the WHO's list of essential medicines (such as tetracycline and doxycycline) are bacterial protein synthesis inhibitors, which mediate their effects by binding ribosomes and halting translation. Although these entities have been of clinical importance for many decades, notable complications are known to arise due to therapy. For example, tetracyclines (which bind both 30S and 50S bacterial ribosomal subunits and are used to treat plague, brucellosis, and Lyme disease) are noted for nephro- and hepato-toxicity, together with anti-inflammatory effects that have shown benefits in patients with chronic

inflammatory skin, autoimmune and neurodegenerative diseases[1,2]. Chloramphenicol - which binds the bacterial 50S ribosomal subunit and is used to treat meningitis, cholera, and typhoid fever - is known to induce bone marrow suppression[3,4]. As antibiotic resistance mechanisms and drugs evolve, so too will the need for efforts to dissect treatment-induced effects on different pathogenic and commensal organisms, as well as host tissues across species.

Mitochondria possess their own genome that encodes core components of the oxidative phosphorylation (OXPHOS) machinery, translated by the organelle's specialized ribosomes, mitoribosomes, in proximity to the inner mitochondrial membrane in order to efficiently generate ATP. Evolutionarily descended from an α-proteobacterial

[1]Department of Urology, The First Affiliated Hospital of Xi'an Jiaotong University, Xi'an, China. [2]Department of Medical Biochemistry and Biophysics, Karolinska Institutet, Stockholm, Sweden. [3]Faculty of Pharmacy, Phenikaa University, Ha Dong, Hanoi, Vietnam. [4]Department of Laboratory Medicine, Karolinska Institutet, Huddinge, Sweden. [5]Department of Microbiology, Tumor and Cell Biology, Karolinska Institutet, Stockholm, Sweden. [6]ARNA Laboratory, Univ. Bordeaux, Centre National de la Recherche Scientifique, Institut National de la Santé et de la Recherche Médicale, Bordeaux, France. [7]Department of Animal and Veterinary Sciences, Aarhus Universitet, Tjele, Denmark. [8]These authors contributed equally: Qiuya Shao, Anas Khawaja, Minh Duc Nguyen. ✉e-mail: xcd@anivet.au.dk; joanna.rorbach@ki.se

ancestor, mitoribosomes share several features with their bacterial counterparts, and it has been long recognized that bacterial protein synthesis inhibitors can interfere with mitochondrial translation and aerobic metabolism, although the molecular bases of such effects have only been described in a handful of cases.

Unlike interactions between bacterial or eukaryotic cytosolic ribosomes and antibiotics, which have been well characterized at near-atomic resolution, only more recently have insights emerged concerning the mitoribosome. For example, streptomycin was shown to bind the mitoribosomal small subunit (mtSSU)[5], while dalfopristin/quinupristin (Q/D) binds the large subunit (mt-LSU), effectively suppressing glioblastoma stem cell growth[6]. The interactions between these antibiotics and mitoribosomes closely resemble that of bacterial ribosome-antibiotic interactions.

In recent years, the multifaceted role mitochondria play in the immune system has become increasingly clear. Via dynamic rearrangements in specialized immune cell states and lineages, the organelle determines effector function and fate via metabolic reprogramming and innate signaling[7,8]. Naive lymphocytes fundamentally depend upon OXPHOS to power clonal expansion[9–12], which if inhibited can affect the effector-memory response[6–9]. Such inhibition could explain beneficial effects of mitochondria-targeting drugs in some clinical contexts[13,14], such as autoimmunity[15–17], although these possibilities require exploration in mechanistic studies.

In this work, we explored the effects of several important antibiotics targeting protein synthesis on human T cells. For the most potent inhibitor of mitochondrial translation and proliferation identified, tigecycline, we then sought to determine the mechanism of action, guiding the design of tetracyclines with reduced off-target binding.

## Results

### Tetracyclines compromise lymphocyte survival in vitro

We first compared the survival of immortalized Jurkat T cells and HeLa cells in the presence of chloramphenicol or doxycycline, two important bacterial ribosome-targeting antibiotics associated with white blood cell phenotypes[4,16,18–22]. We found chloramphenicol to reduce the survival of both cell lines after 3 and 5 days of treatment (Jurkat IC50 = 25.85 μM; HeLa IC50 = 41.71 μM, after 5 days). However, Jurkat T cells were uniquely sensitive to lower concentrations of doxycycline (IC50 = 6.93 μM after 5 days) (Supplementary Fig. 1A).

Tetracyclines (such as doxycycline) are characterized by a four-ring core structure[23]. Their mechanism of action involves reversible binding to the 30S bacterial ribosomal subunit, thereby preventing the attachment of aminoacyl-tRNA to the mRNA-ribosome complex[1]. Given our results, we chose to investigate six additional tetracyclines, including newer derivates reserved for difficult to treat infections (medocycline, methacycline, minocycline, oxycycline, tetracycline and tigecycline). In both Jurkat T cells and peripheral blood mononuclear cells (PBMCs) isolated from anonymized healthy blood donor samples, we found the third-generation tigecycline to have the greatest negative effect on cell survival (IC50 = 2.94–3.08 μM for Jurkat T cells; 2.02−9.42 μM for PMBCs) after 3 days of treatment (Fig. 1A). Tigecycline is a glycylcycline with a N, N-dimethyglycylamido (DMG) moiety attached to position 9 of tetracycline ring D that confers enhanced activity against tetracycline-resistant bacteria.

We then compared the effect of tigecycline to an additional set of widely used bacterial protein synthesis inhibitors (spanning different drug classes - dalfopristin/quinupristin (Q/D), azithromycin, tiamulin, linezolid, and clindamycin (Fig. 1B). In this comparison, tigecycline again showed the greatest toxicity to PBMCs, followed by Q/D, which was previously reported to bind the mitochondrial large subunit[6].

Given our prior observations with doxycycline, it was interesting to observe that HeLa cells were also relatively resistant to 10 μM tigecycline, although Q/D did have a potent effect on their survival

(IC50 = 16.66 μM) (Supplementary Fig. 1B). Q/D also had a greater influence on Hek293 cell survival than did tigecycline and doxycycline (Supplementary Fig. 1B), further illustrating that different cell lines have differential susceptibility to the same compounds.

### Tigecycline inhibits oxidative phosphorylation and primary T cell expansion

To determine whether tigecycline affected mitoribosome function, we profiled mitochondrial and cytosolic translation with/without antibiotic treatment in Jurkat T cells. Metabolic labelling using $^{35}$S-methionine was performed after 18 h of treatment with tigecycline; doxycycline and Q/D served as comparators. We found tigecycline and Q/D inhibited mitochondrial translation at 5 μM, while weaker inhibition was observed for doxycycline (Fig. 1C). On the other hand, 10 μM of these compounds did not have profound effects on cytosolic translation, suggesting mitochondrial protein synthesis is more greatly affected than cytosolic in these rapidly dividing cells (Fig. 1C).

Inhibition of mitochondrial translation by tigecyline was confirmed in primary human T cells using the Mitochondrial Fluorescent Non-Canonical Amino acid Tagging combined with Fluorescence-Activated Cell Sorting (Mito-FUNCAT-FACS) assay, which uses non-canonical amino acid incorporation and click chemistry to measure the rate of translation[24]. In TCR-stimulated CD4+ and CD8+ T cells, we observed dose-dependent inhibition of mitochondrial translation after treatment (Supplementary Fig. 1C, D). Despite the pronounced inhibition in mitochondrial translation, the surviving cells retained comparable mitochondrial mass after 24 h or 7 days of treatment as measured by MitoTracker Green staining (Supplementary Fig. 1E-F).

The defect in translation could be confirmed at the protein level. OXPHOS subunit profiling of Jurkat T cells and human PBMCs by western blot revealed the levels of subunits of complex I (NDUFB8), complex III (UQCRCII) and complex IV (COX2), which contain mitochondrially-encoded molecules, to drop in response to 5 μM tigecycline (Fig. 1D). In contrast, subunits of complex II (SDHB) and complex V (ATP5A) were less affected, indicating treatment-induced nuclear-mitochondrial protein imbalance in the electron transport chain.

Given the pivotal role OXPHOS plays in T cell clonal expansion, we sought to determine whether these deficits correlated with reduced OXPHOS capacity at the cellular level. To do this, we activated T cells within PBMCs via plate-bound anti-CD3/CD28 antibodies and analyzed oxygen consumption using the Seahorse MitoStress test 6 days after stimulation. 5 and 10 μM tigecycline reduced basal, ATP-coupled, maximal and spare oxygen consumption rates (Fig. 1E and Supplementary Fig. 1G). We did not observe a major compensatory increase in glycolysis under these conditions, although our assays were not designed to address acute shifts in fuel usage at different doses (Supplementary Fig. 1H). In the shorter term (18 h post-priming), tigecycline reduced expression of the T cell activation marker, CD25 (IL2RA), by stimulated CD4+ and CD8+ T cells, suggesting cells are affected prior to cell division (Fig. 1F).

To determine whether treatment inhibited primary T cell expansion, we assessed proliferation using flow cytometry. After 6 days of anti-CD3/CD28 stimulation in the presence of IL-2, both CD4+ and CD8+ T cells within healthy donor PBMCs showed dose-dependent reductions in proliferation in the presence of tigecycline (Fig. 1G, H). We also found tigecycline to significantly reduce T cell culture proliferation when administered 24 or 72 h after TCR stimulation, indicating anti-proliferative effects are independent of initial priming (Supplementary Fig. 2A, B). The effects on T cell proliferation by other antibiotics included in the study are presented in Supplementary Fig. 2C.

Given our observations in cell lines (Supplementary Fig. 1B) and previous research indicating differential susceptibility to antibiotics by specialized T cell subsets[15,16], we FACS-isolated naïve (CD45RA+CD27+), central memory (CD45RA-CD27+) and effector memory (CD45RA-CD27-)

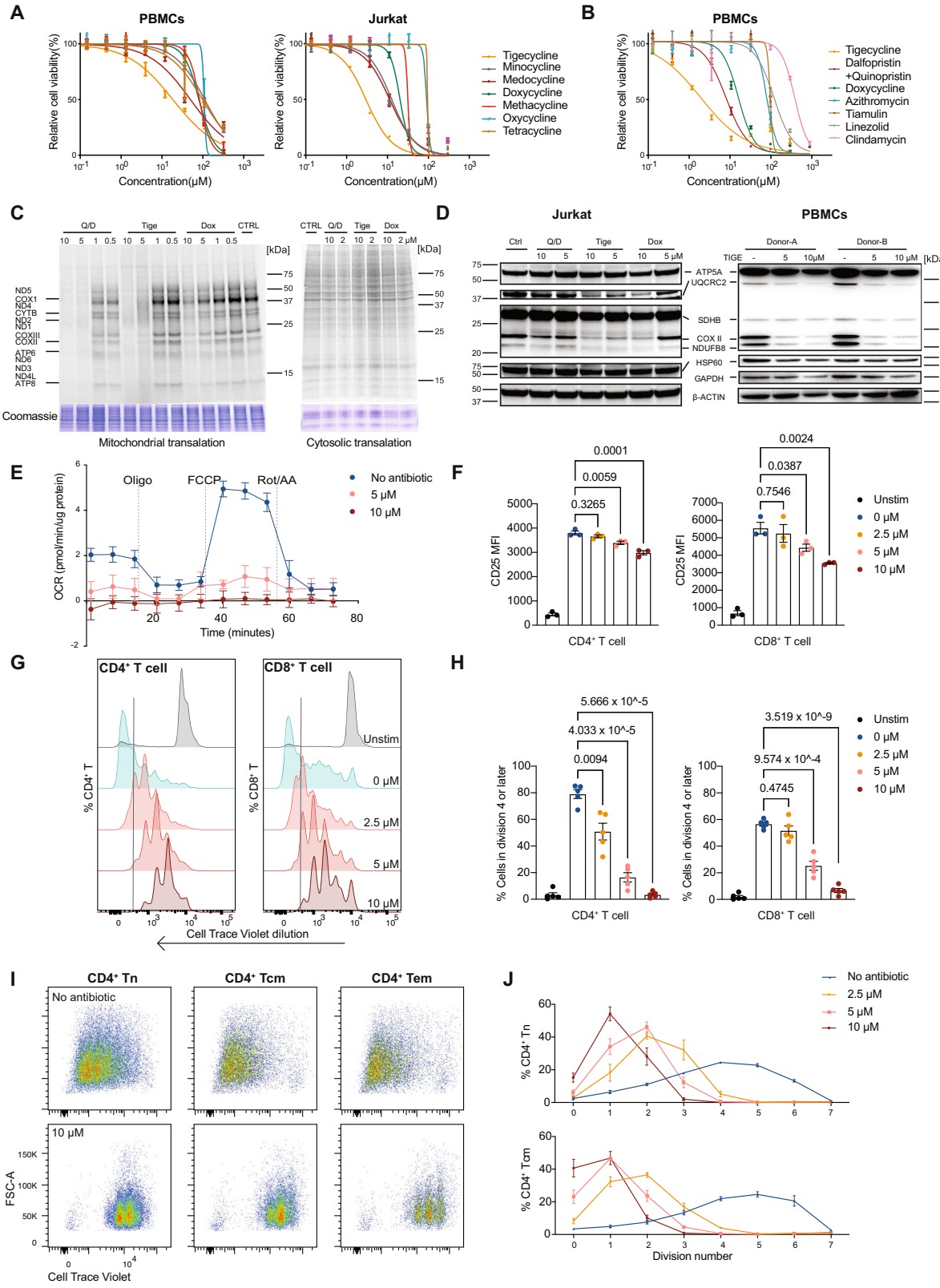

CD4+ T cells from healthy blood donor samples and compared proliferation in the presence or absence of tigecycline. As observed for total T cells, 10 μM tigecyline significantly inhibited the proliferation of all subsets in vitro, with a dose-dependent effect observed in naïve and central memory cells between 2.5 and 10 μM (Fig. 1I, J and Supplementary Fig. 2E−G).

## Structure determination of tigecycline-mitoribosome complexes

To explore the molecular basis of the off-target effects of tigecycline, we isolated 55S mitochondrial ribosomes (mitoribosomes) from Jurkat T cells, incubated them with tigecycline, and analyzed the resulting complexes using single-particle cryo-EM (see "Methods").

**Fig. 1 | Tigecycline compromises human T cell activation and proliferation by inhibiting mitochondrial translation. A** Representative dose-response curves of tetracycline antibiotic cytotoxicity towards PBMCs (left; $n = 5$) and Jurkat T cells (right; $n = 3$) measured 72 h after treatment; mean ± SEM is presented. Data are normalized to DMSO treatment without antibiotics. **B** Representative dose-response curves for additional bacterial protein synthesis inhibitors on PBMCs for 72 h. $n = 6$ biological replicates; mean ± SEM. Data are shown relative to DMSO treatment without antibiotics. **C** $^{35}S$ metabolic labeling assay of mitochondrial (left) and cytosolic (right) translation in Jurkat T cells after 18 h treatment with Q/D, tigecycline, or doxycycline. One representative result from $n = 3$ is shown. **D** Western blot OXPHOS profiling in Jurkat T cells after treatment with Q/D, tige- cycline, and doxycycline and PBMCs after treatment with tigecycline for 6 days. $n = 2$ biological replicates, one shown. **E** Seahorse Mito Stress Test of PBMCs cul- tured with or without tigecycline for 6 days after stimulation with anti-CD3/CD28 antibodies + IL-2 (10 ng/ml). OCR is reported as picomoles (pmol) of $O_2$ per min and normalized according to protein amount/well. $n = 4$ biological replicates, mean ± SEM. Oligo, oligomycin; FCCP, carbonyl cyanide-p-tri-fluoromethoxyphenylhy- drazone; Rot, rotenone; AA, antimycin A. **F** CD25 expression (MFI) on CD4$^+$ and CD8$^+$ T cells 18 h after anti-CD3/CD28 activation + IL-2 (10 ng/ml) in the presence or absence of tigecycline. $n = 3$ biological replicates, mean ± SEM. An RM one-way ANOVA with Tukey's multiple comparisons test was used to analyze the data. **G** T cell proliferation assessed by flow cytometry. PBMC T cells were stimulated with anti-CD3/CD28 beads + IL-2 (10 ng/ml) in the presence or absence of tigecycline. Proliferation was assessed by CTV dilution in live cells after 6 days ($n = 5$ blood donor PBMC samples). PBMCs were labelled with CTV prior to stimulation. **H** Percentage of live T cells in division four or later after antibiotic treatment; data from (**G**), mean ± SEM. An RM one-way ANOVA with Tukey's multiple comparisons test was applied to analyze the data. **I** Proliferation of FACS-isolated CD45RA$^+$CD27$^+$ naïve, CD45RA$^-$CD27$^+$ central memory, and CD45RA$^+$CD27$^-$ effector memory CD4$^+$ T cells from healthy blood donors after 6 days of stimulation with anti-CD3/CD28 beads and IL-2 (10 ng/ml). FACS-isolated cells were seeded in the presence or absence of tigecycline (from 2.5 to 10 μM) and proliferation assessed by CTV dilution in live cells ($n = 3$ blood donor PBMC samples). Representative dot plots from non-treated and treated samples are shown. The FACS gating strategy is shown in Supplementary Fig. 2G. **J** Percentage of live CD4$^+$ T naïve cells and central memory cells in each division; data from (**I**); mean ± SEM. Tn: naïve T cells; Tcm: central memory T cells; Tem: effector memory T cells. Source data for all panels are provided as a Source Data file.

The initial reconstruction of the mitoribosome-tigecycline com- plex yielded an initial cryo-EM density map with an overall resolution of 2.4 Å (Supplementary Figs. 3–5 and Supplementary Table 1). The particles underwent 3D classification with a solvent mask to get rid of poorly aligned particles, which yielded three major classes of mono- somes: (class 1) mitoribosomes with no tRNA ('empty class'), (class 2) with tRNA in the P-site only, (class 3) with A- and P- tRNAs (Supple- mentary Fig. 3). Further local refinements led to a resolution of 2.2 Å for the core region of the monosome (Supplementary Fig. 3 and Supplementary Table 1), detecting all known methylations of 12S and 16S rRNAs, along with 2 iron-sulphur (2Fe-S) clusters in the mtSSU, and 1 Fe-S in the mtLSU, previously identified in the mitoribosomes isolated from Hek293 cells[5]. Interestingly, other reported cofactors of the mitoribosome such as NAD, spermine, and spermidine were not found in any of the classes of tigecycline-bound monosomes. The absence of these cofactors might result from the preferential occu- pancy of these cofactors in specific cell types (T cells versus embryonic kidney cells).

Third-generation tetracyclines are based on a common naphthacene-carboxamide core comprising four rings (A, B, C, and D). The optimization of tigecycline consists of structural modifications of ring D at positions C7 (dimethyl-amino group) and C9 (tert- butylglycylamide)[25] (Fig. 2A). Our analyses unambiguously identified three densities corresponding to tigecycline molecules (Supplementary Fig. 5B–D); one on the mtSSU ('mtSSU site') and two on the mtLSU (mtLSU site-1 and site-2) (Fig. 2B–E). This contrasts to previous struc- tural data from bacteria, which reported a single tigecycline binding site on the SSU[26–28]. The antibiotic's relative occupancy in the three sites was notably high, particularly in the mtSSU and mtLSU site-1, where the P- site-only class of mitoribosomes showed an occupancy exceeding 80% (Supplementary Fig. 5E).

**Tigecycline blocks aminoacyl-tRNA binding to the mtSSU A site**
Tigecycline density on the mtSSU was detected in two mitoribosomal classes, 'P-site tRNA' and 'empty monosome', but not in the class containing A-site tRNA. Based on bacterial studies, it is established that tigecycline specifically targets the head region of the SSU and com- petes with the binding of A-site tRNA, in a similar manner observed for other tetracycline derivatives[27].

The interaction between tigecycline and the mtSSU is analogous to how tetracyclines bind to the small ribosomal subunits of various bacterial species (Fig. 3A–C)[26–33]. In the mtSSU, tigecycline interacts with helix 34 of the 12S rRNA through its polar edge and through ring A, which establish potential hydrogen bonds with the sugar-phosphate backbone of several nucleotides (A1258, U1259, U1325, A1326, G1327,

and G1328) or make indirect contact via coordinated Mg$^{2+}$ ions (G1328) (Fig. 2C and Supplementary Fig. 5B). Ring D stacks on top of nucleo- base U1259, which further stacks on A1326, stabilizing the binding of tigecycline to the mtSSU A-site. Tigecycline coordinates a Mg$^{2+}$ ion (denoted as Mg-1) via its keto-enol system (C11 and C12) and a second one (denoted as Mg-2) via hydroxyl and amide groups of ring A (O3 and O21, respectively). This coordination of the two Mg$^{2+}$ ions is a conserved feature of the tetracycline-ribosome interaction (Fig. 3A–C). We detected the density for an additional Mg$^{2+}$ ion (Mg-3) in close proximity to tigecycline, likely interacting with the oxygen atom from the 9-t-butylglycylamido moiety adopting an extended conformation. Furthermore, the sugar phosphate of U1259 is within hydrogen- bonding distance from nitrogen atoms of the C9 butylglycylamido substituent (Fig. 3A). These extensive interactions stabilize the overall binding of tigecycline to the decoding center of mtSSU. Overall, the interactions observed for the tigecycline-bound mtSSU are compar- able to tetracycline binding with the bacterial SSU, since the functional core of the ribosome including the decoding center is universally conserved[34].

**A unique tigecycline binding site on helix 71 of the mtLSU**
In all our mitoribosome classes, we observed additional density in the region of mtLSU helices 69 and 71 (denoted as mtLSU site-1). We identified tigecycline in this density and observed multiple stabilizing interactions of the antibiotic with the residues forming its binding pocket (Fig. 4A and Supplementary Fig. 5C). In this position, the polar edge of the tigecycline molecule is directed towards the 16S rRNA where it forms multiple interactions, while its C7 extension is near the acceptor stem of the P-tRNA (Fig. 4A).

To form a tigecycline binding site that consists primarily of a hydrophobic cavity between the base of U2628 and the nucleoside of A2604, helix 71 must undergo large conformational changes (Fig. 4A–C). The rearrangement of nucleotides U2626 and U2628 contributes to opening-up a space where the drug can bind. Nucleo- tide U2626 shifts outward and forms stacking interactions with A2598 and C2625. U2628 changes conformation such that its nucleobase becomes intercalated between ring D of tigecycline and the nucleo- base of A2629. These conformational changes are accompanied by a downward shift of helix 71 ( ~4-5 Å) as compared to the untreated monosome (Fig. 4B)[35]. Additionally, nucleotide C2603 flips inward together with A2600, with which it forms stacking interactions (Fig. 4C). This rearrangement positions the nucleobase of A2604 such that it stacks against ring C of tigecycline. Overall, stacking interactions between rings C and D of tigecycline and residues U2628 and A2604 of the 16S rRNA help to stabilize the binding of the drug to this region

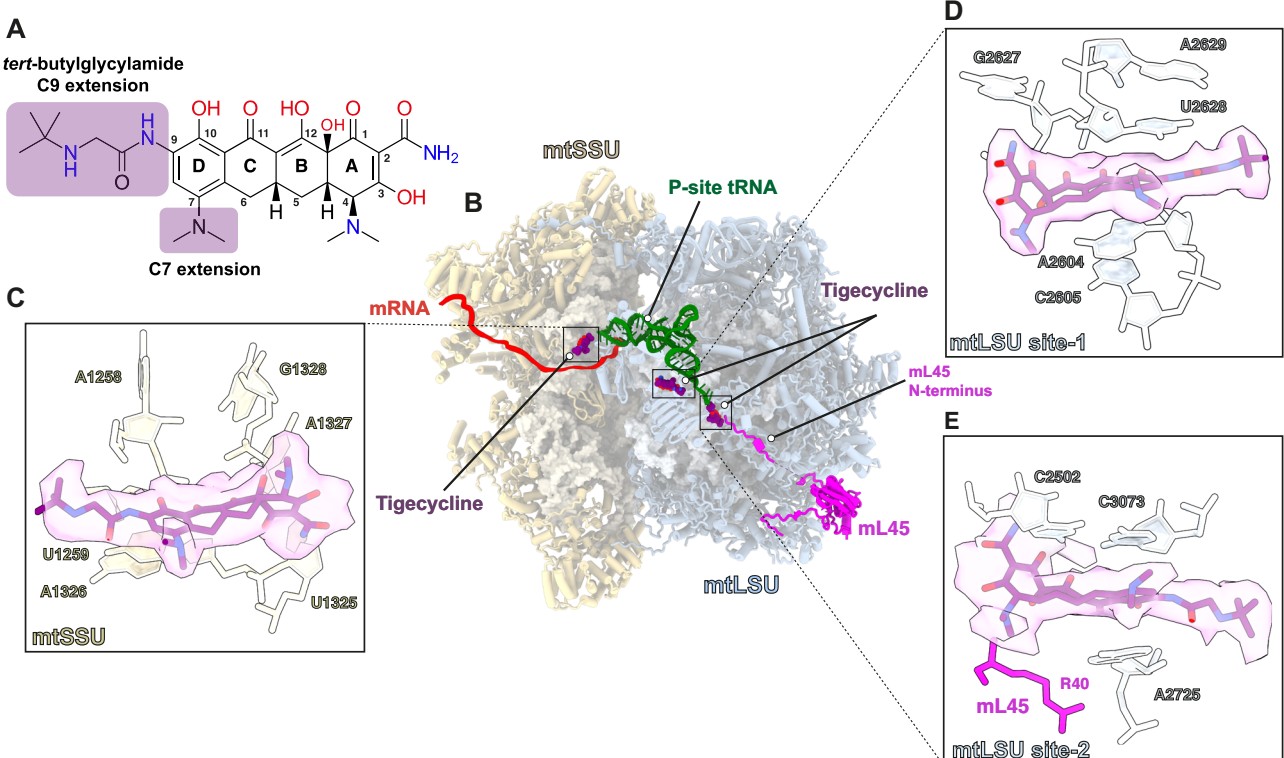

**Fig. 2 | Overview of the interactions of tigecycline with the human mitoribosome. A** Chemical structure of tigecycline. **B** Overview of the three tigecycline (purple) binding sites in complex with the human mitoribosome comprising the mtSSU (yellow), mtLSU (blue), mRNA (red), and P-site tRNA (green). The mL45 (magenta) is depicted with its N-terminal extension occupying the exit tunnel. **C** mtSSU site: tigecycline overlaps with the A-tRNA binding site and is stabilized through multiple interactions. The residues from 12S rRNA which makes direct contact with tigecycline are shown. **D** mtLSU site-1: a novel tigecycline binding site located at helix 71 of the 16S rRNA, close to the acceptor stem of P-tRNA. **E** mtLSU site-2: tigecycline is positioned in the PTC where it stacks against the neighboring nucleotides and makes potential contact with mL45 N-terminus. Carved EM density is shown for tigecycline.

(Supplementary Fig. 5C). Moreover, these conformational changes result in the formation of a non-canonical U-U pair between nucleotides 2599 and 2606, which further contributes to stabilizing the drug-bound conformation of helix 71 (Fig. 4A).

As observed for tigecycline in the mtSSU, the molecule bound to mtLSU site-1 forms metal ion complexes with the phenol-ketone system of rings B and C, and via the oxygen atom of ring A's amide group and the hydroxyl oxygen at position C3. Interestingly, similarly to the mtSSU, we located an additional $Mg^{2+}$ ion density, which coordinates with the hydroxyl of ring D and facilitates indirect interaction with the backbone phosphate of A2604 (Fig. 2D and Supplementary Fig. 5C). Through its polar edge, tigecycline forms potential hydrogen bonds with the phosphate groups of adjacent nucleotide G2627 of 16S rRNA and indirectly through $Mg^{2+}$ ion coordination. The 9-t-butylglycylamido moiety of tigecycline extends towards the PTC center of the mitoribosome, causing nucleotide A3089 to shift (Fig. 4C).

As of now, there is no documentation of antibiotics binding to a comparable location within the bacterial LSU. Interestingly, helix 71 from multiple bacterial species comprises conserved methylations of 23S rRNA surrounding the region where tigecycline binds to the 16S rRNA of the mitoribosome (Supplementary Fig. 6A). In *E. coli* and *T. thermophilus* these correspond to positions $m^5U1939$ and $m^5C1962$, and in *C. acnes*, positions $m^5U2122$ and $m^5C2145$. There is additional methylation of cytosine $m^5C1942$ in *T. thermophilus* (Supplementary Fig. 6A). In contrast, these modifications are absent in the mitochondrial LSU's rRNA. The methylations in helix 71 of 23S rRNA in bacteria are not expected to directly clash with tigecycline; nevertheless, they may confer rigidity of the region, preventing antibiotic binding. Especially, $m^5C1942$ is likely to prevent the conformational change

needed for U1940 to flip out and the drug to bind to the bacterial ribosome (Fig. 4D).

To understand the potential consequences of the binding of tigecycline to the mtLSU site-1 on mitochondrial translation, we compared our structure with previously reported structures of translating mitoribosome. As depicted in Fig. 4E, tigecycline binding may hinder the interaction between the mitoribosome and ribosomal recycling factor 1 (RRF1)[36,37]. This obstruction may impede mitoribosome recycling, resulting in non-functional mitoribosomes. Furthermore, the mtLSU site-1 is positioned between the P and A site tRNA, and binding of the drug to this site may interfere with the translocation of a tRNA from the A site to the P site during the elongation phase of translation. Indeed, the dimethylamine group at C4 of tigecycline might interfere with the phosphate backbone of C71, as well as the OH group of the ribose ring of C70 in the hybrid (A/P site) tRNA (Supplementary Fig. 6B).

## Tigecycline binds to the peptidyl transferase center of the mtLSU

The third density corresponding to a tigecycline molecule was identified at the catalytic peptidyl transferase center (PTC). There have been previous reports of tetracycline-derivative such as sarecycline, anthracycline-like tetracenomycin X (TcmX), as well as the macrolide erythromycin, binding to the PTC and mediating translation arrest in bacteria[30,31,38–40]. Though the occupancy of tigecycline at the PTC in our study is the lowest, as compared to the other two sites in the mtLSU and mtSSU (Supplementary Fig. 5E), it represents a unique occurrence for tigecycline (Fig. 2E). In this position, the ring A of tigecycline faces the lumen of the nascent polypeptide exit tunnel while the 9-t-butylglycylamido extension is in proximity of the terminal adenosine

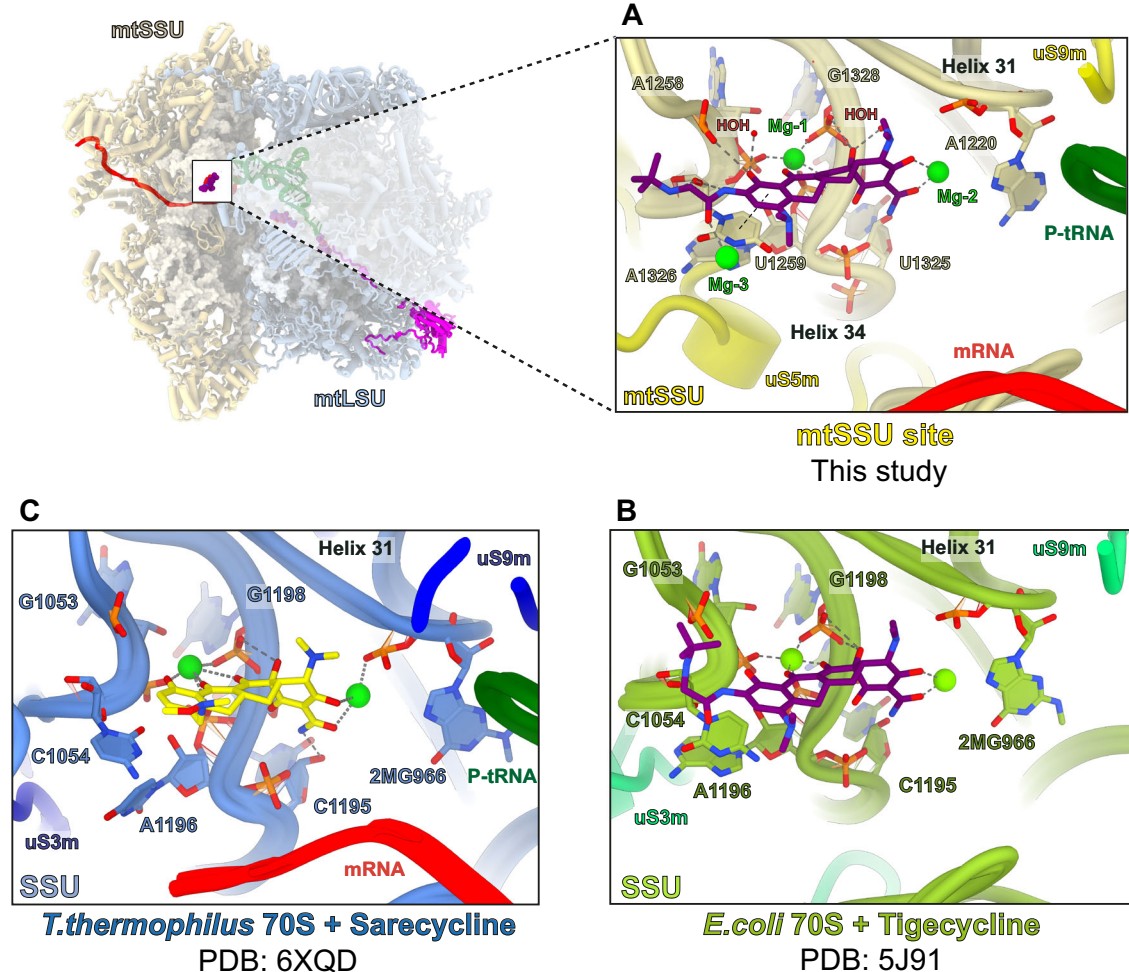

**Fig. 3 | The conserved binding site of tigecycline on the mtSSU. A** Tigecycline establishes canonical interactions with the 12S rRNA of the mtSSU accompanied by coordinated Mg$^{2+}$ ions (lime) and waters (HOH) (red). Hydrogen bond interactions are indicated as dashed lines. **B** Structure of tigecycline (purple) (PDB: 5J91[32]) (**C**) and sarecycline (yellow) (PDB: 6XQD[31]), a tetracycline-derivative, in complex with the SSU of 70S ribosome from *E.coli* and *T. thermophilis*, respectively.

A76 of the P-site tRNA, potentially hindering the progression of the elongating nascent chain along the tunnel (Fig. 5). As observed for the mtSSU and mtLSU site-1, tigecycline mediates canonical metal Mg$^{2+}$ ion complexes through its polar groups at rings B and C, and via the oxygen atoms of the carboxamide group and C3. Further stabilization of tigecycline at the PTC occurs via nucleobase C3073 which stacks on ring D and forms a non-canonical base-pair with C2502 (Fig. 2E and Supplementary Fig. 5D). In the PTC of *C. acnes* 70S, similar non-canonical C1965-C2768 base pairing is observed, contributing to sarecycline binding, whereas corresponding U-U (U1782-U2586) base-pairing in the *E. coli* 23S rRNA is proposed to be crucial for TcmX accommodation (Supplementary Fig. 7)[30,38]. In the absence of a nascent chain, the exit tunnel in mitoribosomes is occupied by the mito-specific N-terminal extension of mL45[41]. A2725 interacts with either the mL45 N-terminal extension or the nascent chain in the exit tunnel, which may contribute to the stabilization of the mL45 or the nascent chain[42]. However, in the tigecycline-bound mtLSU, the A2725 nucleobase is rotated 90° and stacks onto ring D, further securing the binding of tigecycline to the PTC (Fig. 5). The binding sites for tigecycline and sarecycline at the PTC overlap, but surprisingly, sarecycline is rotated 180° and shifted laterally relative to the tigecycline, resulting in its polar edge arranged in the opposite direction (Supplementary Fig. 7). These differences can be either species-specific or antibiotic-specific, requiring further investigations. Ring A of tigecycline is placed away from the N-terminal extension of mL45 in the exit tunnel, and it

coordinates with the Mg$^{2+}$ ion through the oxygen atom of the carboxamide group and hydroxyl group at C3 (Fig. 5 and Supplementary Fig. 5D). Additionally, the oxygen atom at C1 of ring A forms a hydrogen bond with C2502, which forms a *cis* Watson-Crick/Watson-Crick base-pair with C3073 and stacks against the ring D (Fig. 5). The overall positioning of tigecycline allows 9-t-butylglycylamido moiety to extend towards the acceptor arm of the P-tRNA.

## Discussion

In this study, we investigated the off-target effects of bacterial ribosome-binding antibiotics on OXPHOS-dependent lymphocytes. Our screening pinpointed tigecycline as a molecule exhibiting toxicity towards human T cell lines and primary cells, although other members of this class (such as doxycycline) showed similar phenotypes at the cellular level.

While our experiments were not designed to assess antibiotic toxicity in a comparative manner between bacteria and eukaryotic cells, the observation of three binding sites for tigecycline on the mitochondrial ribosome, as opposed to one on the bacterial ribosome, implies that this antibiotic is a potent inhibitor of mitochondrial translation. Extensive studies in bacterial systems have shown that high concentrations of antibiotics may result in secondary binding sites in in vitro experiments, which may not be physiologically relevant[43]. To avoid this issue, we employed a notably lower concentration of tigecycline (30 μM) than that previously used to

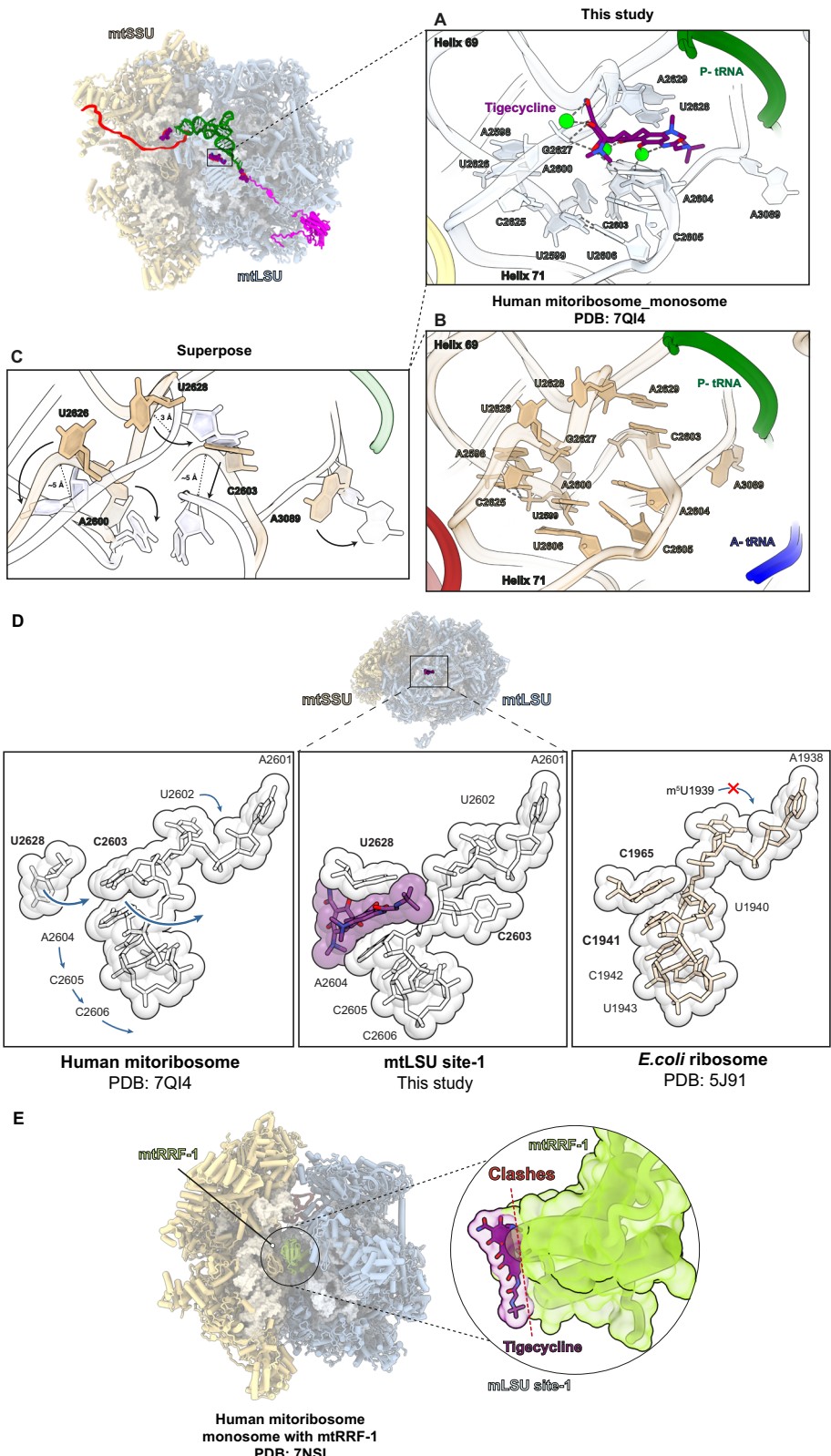

**Fig. 4 | Interaction of tigecycline with helix 71 of the mtLSU (mtLSU site-1) and its functional implications. A** Tigecycline stabilizes in the binding pocket through multiple interactions either directly with the neighbouring nucleotides or indirectly through Mg$^{2+}$ ion coordination as compared to the non-treated mitoribosomes (PDB:7QI4[35]) (**B**). **C** The superposition shows the large displacement of H71 upon the binding of tigecycline. **D** The movement of nucleotide to create a cavity for the binding of tigecycline. The direction of conformational changes of nucleotides occurring upon binding of the drug are shown by arrows. Nucleotides, C2603 and U2628, undergo a large conformational change (bold arrows), while A2604, C2605 and C2606 move downward (smaller arrows) as compared to the non-bound mitoribosome (PDB: 7QI4) resulting in a binding site for the antibiotic. In *E. coli* (PDB: 5J91[32]), the flexible movement of this region is likely restricted by the presence of methylation of U1939. **E** The binding of tigecycline to the mtLSU site-1 might inhibit mitoribosome recycling. The RRF1-bound state (PDB: 7NSI[37]) was superimposed on the tigecycline-bound mitoribosome (this study). The insertion represents a close view at mtLSU site-1 to show the clash of the drug (purple stick and transparent grey sphere) with the RRF1 protein (green).

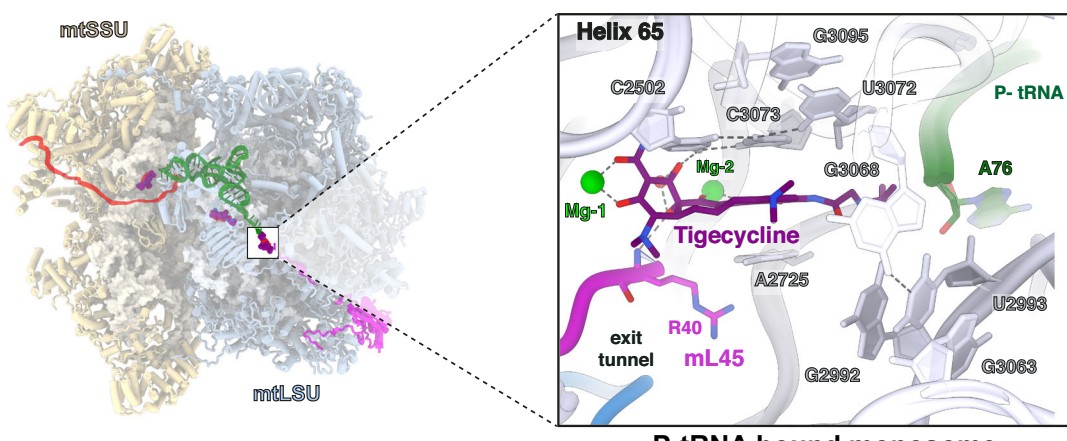

**P-tRNA bound monosome**
This study

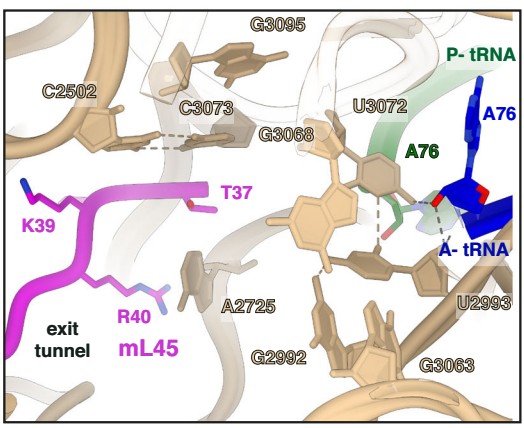

**Human mitoribosome_monosome**
PDB: 7QI4

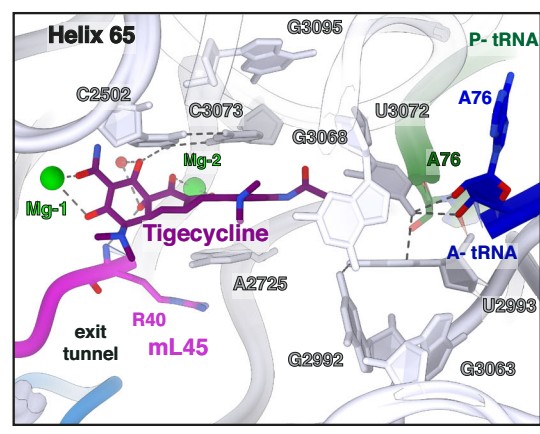

**A- and P-tRNA bound monosome**
This study

**Fig. 5 | Interactions of tigecycline with the PTC (mtLSU site-2).** Tigecycline occupies the PTC, with its 9-t-butylglycylamido substituent stretching adjacent to the P-tRNA (green), and the A ring interacting with the mL45 N-terminal extension (pink). Upon tigecycline binding, A2725 nucleobase shifts and stacks against ring D in both the P-tRNA only and the A- (blue), P-tRNA mitoribosomal classes, as compared to the untreated (tigecycline) mitoribosome (PDB:7QI4[35]). G2992, U2993, and G3063 in the PTC rearranges upon binding of the A-tRNA (as also observed for the untreated mitoribosomes).

investigate its interactions with ribosomes isolated from various bacterial species (concentrations ranging from ~60 to 300 μM)[26–28]. Nevertheless, unlike in previous studies on bacterial ribosomes, where the tigecycline molecule was located only on the SSU, we found two additional binding sites on the mtLSU (Fig. 2), with relatively high occupancy for identified new sites (Supplementary Fig. 5E). The binding to the PTC (mtLSU site-2) overlapped with the recently identified binding site for tetracenomycin X (TcmX) and sarecycline, both with a very similar architecture to tigecycline (Supplementary Fig. 7).

Increasing evidence indicates that many ribosome-targeting antibiotics act in a context-dependent manner, influenced by the nature of the nascent protein. This was recently demonstrated for TcmX, where it was shown to sequester the 3′adenosine of peptidyl-tRNA^Lys in the nascent polypeptide exit tunnel of the ribosome upon translation of a QK motif[40]. Analysis of our tigecycline-PTC site did not detect any changes in the conformation of P-tRNA in either of the mitoribosomal classes comprising P-tRNA only or A- and P-tRNA. This could be due to the absence of the nascent polypeptide in our structures. Unfortunately, current mitoribosome purification methods often do not retain the nascent polypeptide, instead, the exit tunnel in mitoribosomes is occupied by the mito-specific N-terminal extension of mL45, which enters the tunnel once mitoribosomes are detached from the inner mitochondrial membrane. Therefore, the context specificity of tigecycline binding to

the PTC needs to be addressed in the future using other methods, for example, mitoribosome profiling[44].

One of the most intriguing observations arising from our structural studies was the identification of the novel tigecycline binding site on the mtLSU (mtLSU site-1). This specific region has not previously been recognized as a prominent site for antibiotic binding on bacterial ribosomes, raising the question of its mitochondrial specificity. To create sufficient space for the drug to effectively target this area, specific nucleotides including C1963/U2626, C1965/U2628 and U1940/U2603 of helix 71 (*E.coli*/human mtDNA numbering) must undergo substantial movement. However, in bacterial ribosomes, this region contains extensive methylations, constraining the flexibility of modified nucleotides (Supplementary Fig. 6A). In particular, the m5U1939 modification might sterically hinder the conformational change required for accommodating the flipping of U1940 (Fig. 4D). To verify this hypothesis, additional mutagenesis studies are warranted.

Could the binding of tigecycline to mtLSU site-1 contribute to the inhibition of mitochondrial translation? Due to technical limitations, primarily the lack of efficient methods to manipulate the mitochondrial genome and the absence of a robust in vitro mitochondrial translation assay, it is challenging to design an experiment to determine the specific contributions of different binding sites to translation inhibition. However, comparisons with the translating ribosome

suggest that binding to mtLSU site-1 could interfere with mitoribosome recycling and the translocation of tRNA from the A site to the P site (Fig. 4E and Supplementary Fig. 6B). Further studies are needed to confirm these findings. Together, the efficient binding of tigecycline in all three sites is likely to assure an efficient inhibition of several stages of translation.

Our study reports the first structures of mitoribosomes isolated from human lymphoid cells (Jurkat T cells). Given that the features of the binding sites remain conserved with other mammalian mitoribosomes previously characterized (from Hek293 and porcine), it suggests that the interaction is likely to be conserved across mammals. However, our toxicity tests revealed a significantly stronger effect of this antibiotic on Jurkat T cells, compared to Hek293 and Hela cells (Supplementary Fig. 1B). This point is particularly important considering the numerous studies modulating mitochondrial function in the context of cancer[45]. Antibiotics that potentially inhibit mitochondrial translation, including tigecycline, have been reported to inhibit survival in glioblastoma cells[6], leukemia stem cells[46], ovarian cancer cells[47], renal cancer cells[48], and imatinib-resistant chronic myeloid leukemia (CML)[49] cells. Our data encourage the investigation of anti-tumor immunity in patients receiving such treatments but also illustrate different cell types respond differently to OXPHOS inhibition, depending on the genetic program. Indeed, low doses of tetracyclines promoted the survival of fibroblasts derived from mitochondrial disease patients, which may have different compensatory mechanisms to cope with reductions in OXPHOS[22]. Future studies should, therefore, investigate how antibiotics affect different cells in vivo across clinical contexts. In an infectious setting, antibiotics need to be delivered to affected lesions to combat the pathogen. The observed IC50 values for tigecycline against T cells in vitro were relevant to drug concentrations observed in soft tissue lesions of patients undergoing treatment for bacterial infections[49–51]. Therefore, future studies should consider how treatments may influence pathogen clearance (e.g., macrophage function), immunity, and tissue repair.

Together, these data provide a molecular mechanism to explain part of the anti-inflammatory effects of tetracyclines, informing antibiotic and OXPHOS inhibitor design[52].

## Methods

### Ethical declaration
Ethical approval for the use of peripheral blood samples from human donors was granted by the Swedish Ethical Review Authority (registration number 2018/1498-31/3). All human studies were carried out in accordance with the guidelines and policies of Karolinska Institutet and EU legislation.

### Human samples
Buffy coats were ordered from anonymous blood donors at Karolinska Universitetslaboratoriet. On the same day as sample collection, PBMCs were isolated by density gradient centrifugation over Lymphoprep (StemCell Technologies), washed with RPMI-1640 (Cytiva HyClone) with 10% FBS, and cryopreserved at −80 °C in FBS with 10% DMSO (Sigma)[53].

### Cell viability assays
Cells were seeded in 96-well plates ranging from 2000 to 15,000 cells per well depending on different cell lines and treated with antibiotics in serial dilutions for 72 h (120 h for the preliminary assays) in a humidified incubator (37 °C, 5% CO$_2$) in 100 µl medium (DMEM with 10% FBS, 2.05 mM L-glutamine (Sigma) for Hek293 and Hela; RPMI-1640 with 10% FBS, 2.05 mM L-glutamine (Sigma), and 55 µM 2-ME (Gibco) for Jurkat T cells and PBMCs). Based on preliminary tests, the number of cells per well was optimized to 2000 for Hela and Hek293, 6000 for Jurkat T cells, and 15,000 for PBMCs. The stocks of antibiotics were prepared by dissolving in DMSO. As control, the same amount of DMSO was added to the non-treated groups. Serial drug dilutions were prepared in the corresponding medium to provide a

total of 9 concentrations up to 300 µM for tetracyclines and up to 900 µM for additional bacterial protein synthesis inhibitors. After drug treatment, 10% CCK-8 (Cell Counting Kit-8, Sigma Aldrich) (10 µl per well) of the total volume was added to the cells and incubated for 1–4 h depending on cell lines (1 h for Hela and Hek293; 3 h for Jurkat T cells; 4 h for PBMCs) before the evaluation of the effect of antibiotics on cell viability. The number of viable cells was calculated by measuring the absorbance at 450 nm using a microplate reader and normalized to the non-treated control. Results were visually confirmed under the light microscope. Each treatment was performed in technical repeats. Dose-response curves were plotted, mean ± SEM was shown, and half maximal inhibitory concentration (IC50) values were calculated in GraphPad Prism.

### De novo mitochondrial translation assay
Jurkat T cells were seeded in 6-well plates at a density of 1 million cells/ml and cultured with different concentrations of antibiotics for 18 h in RPMI-1640 medium (with 10% FBS, 2.05 mM L-glutamine (Sigma), and 55 µM 2-ME (Gibco)). After centrifugation, cells were incubated twice for 5 min at 37 °C in Cys-/Met-free medium (DMEM, high glucose, no glutamine, no methionine, no cysteine, supplemented with 10% dialyzed fetal bovine serum, 1× GlutaMax, and sodium pyruvate), followed by 20 min incubation at 37 °C in Cys-/Met-free medium supplemented with 100 µg/ml emetine to inhibit cytosolic translation (only for mitochondrial translation groups). Subsequently, 1 ml Cys-/Met-free medium supplemented with 200 (for cytosolic translation) or 400 µCi (for mitochondrial translation) of EasyTag EXPRESS [$^{35}$S] protein labeling mix (methionine and cysteine) (Perkin Elmer) were added to each sample and incubated for 20 min at 37 °C, 5% CO$_2$. Following labeling, cells were washed three times with 1 ml PBS, harvested by centrifugation (400 × $g$, 10 min, 4 °C), and stored at −20 °C. Cells were lysed by resuspension in 30 µl PBS supplemented with a Complete EDTA-free protease inhibitor cocktail and 50 U Benzonase nuclease (Sigma) and by application of one freeze-thaw cycle. Protein contents were determined by Pierce BCA assay (Thermo Fisher). 1 × NuPAGE LDS sample buffer (Thermo Fisher) was added to 30 µg lysate for mitochondrial translation and 10 µg lysate for cytosolic translation, respectively, and separated on NuPAGE 12% Bis-Tris gels (Thermo Fisher). Coomassie staining was performed using Imperial Protein Stain (Thermo Fisher) according to manufacturer suggestions. The gel was fixed in fixing solution (20% methanol, 7% acetic acid, 3% glycerol) for 1 h at RT and vacuum-dried at 65 °C for 2 h. The resultant gel was exposed to storage Phosphor screens (Fujifilm) and visualized with Typhoon FLA 7000 Phosphorimager (GE Healthcare).

### Mito-FUNCAT-FACS
Thawed PBMCs were washed with RPMI (+10% FBS, 2.05 mM L-glutamine (Sigma), and 55 µM 2-ME (Gibco)) before resting at 37 °C (5% CO$_2$) for 30 min. After counting, cells were resuspended in FACS buffer at a density of 5 × 10$^7$ cells/ml. T cells were isolated using EasySep™ Human T Cell Isolation Kit (Stemcell Technologies). Purified T cells were counted and 85,000 cells per well seeded onto 96-well U-bottom plates with/without Human T-Activator CD3/CD28 beads (Invitrogen) at a 1:2.8 cell:bead ratio and 10 ng/ml IL-2 (Proteintech) in RPMI-1640 (+10% FBS, 2.05 mM L-glutamine (Sigma), and 55 µM 2-ME (Gibco)) for 48 h. To assess mitochondrial translation, activated T cells were washed twice with methionine/cysteine/pyruvate-free DMEM and incubated in methionine-depletion medium (DMEM supplemented with 10% dialyzed FBS, 1× GlutaMAX, 1× sodium pyruvate, and 0.2 mM L-cystine) for 30 min at 37 °C. To inhibit cytosolic translation, anisomycin (100 µg/mL) was added to the medium. For mitochondrial translation inhibition, 100 µg/mL chloramphenicol (309 µM, positive control) or tigecycline (2.5–10 µM) were added to the medium. Cells were then labeled with 50 µM L-homopropargylglycine (HPG) for 3 h at 37 °C, during which time the medium still contained antibiotics.

Following HPG labeling, cells were collected and stained with surface markers (CD4, CD8). Cells were pre-permeabilized with 0.005% digitonin in mitochondria-protective buffer (10 mM HEPES/KOH, 10 mM NaCl, 5 mM MgCl2, 300 mM sucrose, pH 7.5) for 3 min, fixed with 4% paraformaldehyde for 10 min, and fully permeabilized with 0.25% Triton X-100 for 5 min. After blocking with 5% BSA for 10 min, incorporated HPG was detected via copper-catalyzed cycloaddition using 0.05 µM Alexa Fluor 488 picolyl azide (Click-iT™ Plus Alexa Fluor™ 488 Picolyl Azide Toolkit, Gibco) for 30 min at room temperature. Subsequently, the reaction cocktail was removed and cells resuspended in FACS buffer. Flow cytometry was carried out on a BD Celesta (BD Biosciences) and analyzed in FlowJo (TreeStar). Data are presented as fold change relative to negative controls.

### Western blotting for OXPHOS complexes

Human PBMCs and Jurkat T cells were seeded in 6-well plates at a density of 1 million cells/ml and incubated with or without antibiotics in RPMI-1640 medium (with 10% FBS, 2.05 mM L-glutamine (Sigma), and 55 µM 2-ME (Gibco)) for 6 days. After centrifugation, cells were lysed in RIPA buffer containing protease and phosphatase inhibitors (cOmplete and PhosSTOP, respectively, Roche). Protein concentrations were determined by BCA (Pierce, Thermo Fisher). Protein samples were resuspended with 1× NuPAGE LDS sample buffer (Thermo Fisher) supplemented with 100 mM dithiothreitol, heated for 10 min at 75 °C, and separated on NuPAGE 4–12% Bis-Tris mini gels (Thermo Fisher) using NuPAGE 1× MES (Thermo Fisher) running buffer, and transferred to PVDF membranes (0.45 µm, Immobilon-P, Sigma) using the iBlot2 system (Invitrogen). Non-specific binding was blocked with TBST containing 5% non-fat milk, followed by overnight incubation at 4 °C with antibodies against Total OXPHOS Human WB Antibody Cocktail (Abcam, Cambridge, UK), followed by HRP-conjugated secondary antibodies for 1 h at room temperature. β-actin, HSP60, and GAPDH (Cell Signaling Technology) were used as loading control. Proteins were detected using Clarity Western ECL Substrate (Bio-Rad, 170-5061). For an example of presentation of full scan blots, see the Source Data file.

### Extracellular metabolic flux assay

The Seahorse XFe96 Analyzer (Seahorse Bioscience) was used to measure oxygen consumption rate (OCR) and extracellular acidification rate (ECAR) in human PBMCs. $1.5 \times 10^6$ cells/well were seeded in 24-well plates and stimulated using plate-bound anti-CD3 antibody (2 µg/ml, clone OKT3, BD Biosciences), soluble anti-CD28 antibody (0.5 µg/ml, clone CD28.2, Biolegend) and IL-2 (10 ng/ml, Proteintech) in RPMI-1640 (with 10% FBS, 2.05 mM L-glutamine (Sigma), and 55 µM 2-ME (Gibco)). Cells were treated with or without 5 and 10 µM Tigecycline for 6 days with a half medium change after 3 days. After treatment, cells were washed with Seahorse XF RPMI assay medium, and then 200,000 cells per well were seeded (in triplicate) in 40 µl assay medium in XF 96-well cell culture microplate coated with poly-D-lysine (Sigma). The plate was centrifuged at $300\,g$ for 5 s with no brake, rotated 180°, and centrifuged again for 5 s at $300 \times g$. After centrifugation, 140 µl of assay medium was added per well and the plate was left to stabilize in a 37 °C, non-CO₂ incubator for 40 min. During seahorse, wells were sequentially injected with compounds to achieve final concentrations of 1264 µM oligomycin (Sigma); 2 µM FCCP (Sigma); 0.5 µM rotenone (Sigma) together with 0.5 µM antimycin A (Sigma). OCR was measured for each well three times, every three min, before and after each injection. OCR was normalized to protein concentration using a BCA Protein Assay kit (ThermoFisher Scientific) conducted according to the manufacturer's instructions.

### In vitro stimulation of human PBMCs

After thawing frozen PBMCs, cells were washed with RPMI complemented with 10% FBS, 2.05 mM L-glutamine (Sigma), and 55 µM

2-ME (Gibco) and rested at 37 °C (5% CO₂) for 30 min. 250,000 cells per well were seeded in 96-well U-bottom plates with/without Human T-Activator CD3/CD28 beads (Invitrogen) at a 1:1 ratio and 10 ng/ml IL-2 (Proteintech) in RPMI-1640 medium mentioned above. At the same time, antibiotics were added. For the short-term activation assay, cells were harvested 18 h post-stimulation and stained with Aqua fixable live/dead dye (Invitrogen) and surface markers. For the proliferation assays, cells were stained with CFSE (eBioscience) or Cell Trace Violet (ThermoFisher Scientific) dyes before seeding and stimulation, according to the manufacturer's recommendations. For TCR priming experiments, tigecycline was added at 0, 24 or 72 h post-activation. For mitochondrial mass measuring, cells were stained with Mitotracker Green Fm Dye (Thermo Fisher Scientific) after incubation with Tigecycline. After incubation for 6 days with a half medium (complete, with antibiotics and IL-2) change after 3 days, cells were harvested via centrifugation, and stained with live/dead dye and surface markers for analysis by flow cytometry. The Foxp3/Transcription Factor Staining Buffer Set (Invitrogen) was used to fix cells according to the manufacturer's instructions. The FACS gating strategy is shown in Supplementary Fig. 2D. Flow cytometry was carried out on a BD Celesta (BD Biosciences) and analyzed in FlowJo (TreeStar) software.

### Human CD4+ T cell sorting and stimulation

Thawed PBMCs were washed with RPMI complemented with 10% FBS, 2.05 mM L-glutamine (Sigma), and 55 µM 2-ME (Gibco) before resting at 37 °C (5% CO₂) for 30 min. After counting, cells were resuspended in FACS buffer at a density of $5 \times 10^7$ cells/ml. CD4+ T cells were isolated using CD4+ T cell negative isolation kits (Miltenyi Biotech) and enriched using CD4+ T cell Enrichment kits (Miltenyi Biotech). Purified CD4+ T cells were counted and stained with CTV dye and live/dead dye. Subsequently, surface makers were stained at 4 °C in FACS buffer (DPBS + 2% FBS). The following panels were used to classify human T CD4+ cells from the live lymphocyte gate: naïve CD4+ T cells: Aqua⁻CD4+CD45RA+CD27+; central memory CD4+ T cells: Aqua⁻CD4+ CD45RA⁻CD27+, effector memory CD4+ T cells: Aqua⁻CD4+ CD45RA⁻CD27⁻. After staining, cells were washed, resuspended and sorted using a FACSAria Fusion (BD Biosciences). Sorted cells were collected into PBS complemented with 2% FBS at 4 °C. Cells were then harvested via centrifugation and resuspended in RPMI-1640 (+10% FBS, 2.05 mM L-glutamine (Sigma), and 55 µM 2-ME (Gibco) before counting. Sorted CD4+ subsets were counted and 85,000 cells per well seeded onto 96-well U-bottom plates with/without Human T-Activator CD3/CD28 beads (Invitrogen) at a 1:2.8 cell:bead ratio and 10 ng/ml IL-2 (Proteintech) in RPMI-1640 (+10% FBS, 2.05 mM L-glutamine (Sigma), and 55 µM 2-ME (Gibco). At the same time, tigecycline in serial concentrations (from 2.5 to 10 µM) was added. A half medium change took place after 3 days, and cells were harvested after 6 days for flow cytometry. Antibodies used in the study are presented in Supplementary Table 2.

### Isolation of mitochondria and purification of mitochondrial monosomes

Jurkat T cells were grown in RPM1-1640 medium (+10% FBS, 2.05 mM L-glutamine (Sigma), and 55 µM 2-ME (Gibco)) in a vented flask shaking at 120 rpm at 37 °C under 5% CO₂. The culture was scaled up by splitting at a cell density of $1.6 \times 10^6$ cells/ml. A final volume of 1.5-liter cells at a density of $1 \times 10^6$ cells/ml was harvested by centrifugation at $1000 \times g$ for 10 min at 4 °C. After being washed with cold PBS buffer, the cell pellet was resuspended in the cold hypotonic MSE buffer (with 0.6 M mannitol, 10 mM Tris–HCl pH 7.4, 1 mM EDTA, 0.1% BSA), and ruptured on ice by a semi-automatic homogenizer (Schuett-biotech). The lysate was clarified by centrifugation at $400 \times g$ and 4 °C for 10 min. The pellet was resuspended and subsequently homogenized. After 3 cycles of homogenization-centrifugation, the cell lysates were combined and the mitochondria were pelleted by additional

centrifugation at 11,000 x g and 4 °C for 10 min. The crude mitochondria were loaded onto the sucrose cushion (1.0 M and 1.5 M sucrose in, 20 mM Tris–HCl pH 7.4, 1 mM EDTA) and centrifuged for 1 h at 77,000 x g (25,000 rpm) in a SW41 Ti rotor (Beckman Coulter). The band formed by the mitochondria in the middle between 1 and 1.5 M sucrose was collected carefully and resuspended in 10 mM Tris–HCl pH 7.4 in a 1:1 ratio, The pure mitochondrial pellet was collected after centrifugation at 11,000 x g and 4 °C for 15 min and then resuspended in mitochondrial freezing buffer (300 mM trehalose, 10 mM Tris–HCl pH 7.4, 10 mM KCl, 0.1% BSA, 1 mM EDTA), flash-frozen in liquid $N_2$ and stored at −80 °C.

The purified mitochondria were thawed and lysed by incubating at 4 °C for 30 min in the lysis buffer (25 mM HEPES–KOH pH 7.5, 20 mM Mg(OAc)$_2$, 50 mM KCl, 2% (vol/vol) Triton X-100, 2 mM Dithiothreitol (DTT), 1× cOmplete EDTA-free protease inhibitor cocktail (Roche), 40 U/μl RNase inhibitor (Invitrogen)). The mitochondrial lysate was centrifuged at 19,000 x g (13,000 rpm) for 12 min at 4 °C, and subsequently overlaid on top of a 10–30% sucrose gradient in the ribosome buffer (25 mM HEPES/KOH pH 7.5, 50 mM KCl, 20 mM Mg(OAc)$_2$, 2 mM DTT). After centrifugation for 21 h at 54,331 x g (21,000 rpm) in a SW41 Ti rotor (Beckman Coulter), the gradients were fractionated with a Biocomp Fractionator. Fractions corresponding to the monosomes were pooled and pelleted at 135,520 x g (55,000 rpm) for 16 h at 4 °C using a TLA55 rotor (Beckman Coulter). The pellet was gently washed 3 times and dissolved in ribosome buffer. The solution was kept on ice for 15 min and the soluble mitoribosomes were collected by centrifugation at 20,000 x *g* for 15 min at 4 °C to get rid of aggregation. The concentration of the purified monosomes was quantified by nanodrop.

### CryoEM data collection and analysis

In vitro reconstitution of the mitoribosomal monosome-tigecycline complex was performed using 30 μM of tigecycline incubated with 100 nM of monosome in a ratio of 1:300. Holey carbon grids (Quantifoil R2/2, copper, 300 mesh) coated with a layer of continuous carbon (~3 nm thickness) were subjected to a glow discharge of 25 mA for 120 s. The sample was applied to the grids at 4 °C with 100 % humidity and incubated for 30 s using a Vitrobot MKIV (Thermo Fisher Scientific), followed by 3 s blotting with blot force 3 and plunge-freezing in liquified ethane. The dataset was acquired on Titan Krios G3i transmission electron microscope (Thermo Fisher Scientific) operated at 300 kV in the Karolinska Institutet's 3D-EM facility using a slit width of 20 eV with GIF quantum energy filter (Gatan). For imaging, a K3 detector (Gatan) was used capturing micrographs at magnification of 165kX yielding a pixel size of 0.505 Å. A dose of 45 electrons per Å² in 50 frames was used with defocus values ranging from −0.4 to −1.6 μm. Motion correction followed by CTF estimation, Fourier cropping (to 1.01 Å/px), particle picking, and extraction in 512-pixel boxes (size threshold 300 Å, distance threshold 20 Å, using the pre-trained Box-Net2Mask_20180918 model) were performed on the fly using Warp[54]. Only particles from micrographs with an estimated resolution of 5 Å were retained for further processing. Detailed parameters are given in Supplementary Table 1.

A total of 12,972 micrographs from 2 datasets were selected based on an estimated resolution cut-off of 5 Å and defocus below 2 μm as estimated by Warp. A total of 519,237 picked particles from Warp were imported to CryoSPARC (v4.2.1)[55] to perform further processing. 2D classification was carried out followed by ab initio reconstructions of clean or good classes with high-resolution features and junk classes. These ab initio reconstructions were used for the heterogeneous refinements of all picked particles. After several rounds of heterogeneous refinements, 266,250 clean particles comprised the mitoribosomal monosome with high-resolution features were retained and used for further processing. Homogenous refinement of these clean particle stacks was performed which yielded a resolution of 2.4 Å. A recent model of human mitoribosomal monosome (PDB:7QI4[35]) purified from the Hek293 T cells bound to A-tRNA, P-tRNA, and mRNA was fitted in our reconstruction of the monosome, which also comprises these features with additional densities in both mtSSU and mtLSU that could be attributed to the antibiotic tigecycline. A mask targeting the P-tRNA was generated to perform 3D variability analysis and subsequently classified into three particle clusters, with each representing different features of the monosome. As a result, a subset of 11,757 particles were excluded after this round due to a poor electron density map. Next, a total of 124,050 particles with clear density for the P-tRNA were subjected to a second round of 3D variability analysis for further classification into three particle clusters using the A-tRNA as a mask. Finally, after two steps of classification by 3D variability analysis, three major classes were obtained, class 1 with the majority of the particles ('Empty': 130,443 particles) lacking densities for the A-site and P-site tRNA, class 2 ('P-tRNA only': 87,076 particles) lacking the occupancy for the A-tRNA but with P-tRNA only and mRNA bound, and class 3 ('A- and P-tRNA': 36,974 particles) contained monosome bound A-tRNA, P-tRNA, and mRNA (Supplementary Fig. 3). Each class/particle set was subjected to a homogenous refinement which yielded a corresponding reconstruction at 2.5 Å, 2.7 Å, and 2.8 Å resolution, respectively. To improve the local resolution of the tigecycline-bound regions on the monosome, 3D refined classes were subjected to CTF refinement (global and local refinement). Furthermore, two masks covering the region of the tigecycline-binding sites on the mtSSU (SSU-head) and mtLSU (LSU-body) were prepared (Supplementary Fig. 3), and local-masked 3D refinement was performed. Reported resolutions are based on gold-standard, applying the 0.143 criterion on the FSC between the reconstructed half-maps (Supplementary Fig. 3-4). Maps underwent local-resolution filtering, superposed to the consensus map, and combined via Phenix[56] for model building and refinement.

### Model building and refinement

Model building of the tigecycline-bound monosome was carried out using *Coot*[57]. The starting model for the monosome was Protein Data Bank (PDB) ID 7QI4[35]. This model was fitted as a rigid body into the map of the 'P-tRNA only' monosome class, and further adjustments were made manually. Three active sites for tigecycline were identified on the monosome, which agreed with the density: one at the mtSSU and two in the mtLSU. Water molecules were picked by *Coot* automatically around the tigecycline-binding region, and adjusted manually. Metal ions, cofactors (2Fe-S), and modifications were placed based on map densities. Geometrical restraints of modified residues and ligands were calculated by Grade Web Server (http://grade.globalphasing.org) or obtained from the CCP4 library[58]. Hydrogens were added to all molecules except water by REFMAC5[59]. The final model was refined using the composite map via Phenix.real_space_refine v1.18[56]. The refined model was validated with MolProbity[60,61] and the Phenix suite[48]. Model statistics are listed in Supplementary Table 1. UCSF ChimeraX 1.6.1[62] was used to make the figures.

### Statistical analyses

Statistical analyses were carried out in Prism v9 (GraphPad). Differences between groups were analyzed by a Student's t-test or one-way ANOVA with Tukey's multiple comparisons test.

### Reporting summary

Further information on research design is available in the Nature Portfolio Reporting Summary linked to this article.

## Data availability

The Cryo-EM maps generated in this study have been deposited at the Electron Microscopy Data Bank as follows: Class 1 (empty class), EMD-19544 (consensus map), EMD-19545 (SSU-head), EMD-19546 (LSU-

body); Class 2 (P-site tRNA), EMD-19493 (consensus map), EMD-19490 (SSU-head), EMD-19491 (LSU-body), EMD-19460 (composite); Class 3 (A- and P-site tRNA), EMD-19526 (consensus map), EMD-19539 (SSU-head), EMD-19542 (LSU-body), EMD-19460 (composite map). Associated molecular model has been deposited at the PDB: 8RRI (tigecycline bound human mitoribosome containing P-site tRNA and mRNA). Source data are provided with this paper.

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

## Acknowledgements

The work was funded by Max Planck Institute-Karolinska Institute, the Knut & Alice Wallenberg Foundation (WAF2017, KAW 2018.0080, to J.R.), Swedish Research Council (VR2022-02179, to J.R.), EMBO (STF 7213, to X.C.D.; LTF 2020-606, to M.D.N.) and Chinese Scholarship Council fellowship (No.202006280090, to Q.S.).

## Author contributions

Methodology: Q.S., A.K., M.N., J.Z., Y.L., J.N., M.A., X.C.D., J.R.; Analyses: Q.S., A.K., M.N., V.S., A.I., X.C.D., J.R.; Writing Original Draft: Q.S., X.C.D., J.R.; Review & Editing: all authors. Visualization: A.K., M.N., V.S., A.I.; Supervision and Funding Acquisition: X.C.D., J.R.; Q.S., A.K., M.N. contributed equally and have the right to list her/himself first in bibliographic documents.

## Funding

## Competing interests

The authors declare no competing interests.
