## [Transparent Peer Review file · Nature Communications]

T cell toxicity induced by tigecycline binding to the mitochondrial ribosome

Corresponding Author: Dr Joanna Rorbach

Version 0:

Reviewer comments:

Reviewer #1

(Remarks to the Author)

Shao et al. (2024) present a comparative study investigating the cytotoxic effects of various antibiotics on T cells, highlighting tigecycline, a third-generation tetracycline, as particularly potent. Their in vitro experiments showed that tigecycline at concentrations of 5 μ M inhibits mitochondrial translation without affecting cytosolic translation. Using cryo-electron microscopy (cryo-EM), they analyzed mitochondrial ribosomes from Jurkat T cells, revealing tigecycline's binding to the A-site of the small ribosomal subunit, preventing A-site tRNA binding, and identifying two additional binding sites in the large mitochondrial ribosomal subunit. These findings align with a recent study by Li et al. (Nat Commun, 2024), which also demonstrated tigecycline binding to both the small and large ribosomal subunits.

Major Points

The originality of Shao et al.'s study is questionable. Li et al.'s recent publication in Nature Communications has already demonstrated that tigecycline binds to both the small and large mitochondrial ribosomal subunits, inhibiting translation. This overlap in findings raises concerns about the additional value provided by Shao et al.'s work. Without further insights, it is unclear how this study will advance the field.

Minor Points

The cryo-EM data processing workflow lacks sufficient detail (Fig S2). The authors should expand on this workflow for reproducibility and transparency. Specifically, including a representative micrograph and the major 2D classes used in their analysis would provide a better understanding of the data quality.

The authors should enhance the labeling of the FSC curves. Additionally, they should also present map-model FSCs.

The manuscript does not specify the number of classes used in each 3D classification round. This information is important for evaluating the thoroughness and robustness of their structural analysis. Indeed, there is a discrepancy between the sum of particles with P site tRNA and empty P site when compared to the total number of particles used for the initial 3D refinement.

In Figure 2B, the authors present an atomic model for mRNA spanning the entire mRNA channel. However, without access to the corresponding map and coordinates, it is impossible to assess whether the map's quality justifies building the mRNA model from the entry to the exit site. The authors should include a map-to-model fitting for mRNA and tRNA as a supplementary figure.

How does the binding site in the SSU compare with that reported by Brodersen et al. 2000 (PMID: 11163189)?

Reviewer #3

(Remarks to the Author)

Shao, Khawaja, Nguyen, and colleagues explore the impact of the antibiotic tigecycline on both immortalized and primary human T cells. Due to the endosymbiotic origins of mitochondria, it is well-established that mitochondrial ribosomes share

several structural and functional similarities with bacterial ribosomes. Building on this understanding, as well as existing literature suggesting that antibiotics adversely affect mitoribosome function, the authors demonstrate that tigecycline compromises T cell survival and proliferation by impairing oxidative phosphorylation (OXPHOS). This is followed by a detailed structural characterization of the specific binding sites of tigecycline on mitoribosomes, employing single-particle cryo-electron microscopy. The results obtained explain the inhibitory effect of tigecycline on mitochondrial translation. While other recent studies have examined tigecycline's interaction with mitoribosomes, the significance of this work is preserved by its focus on immune cells. Further characterization of mitochondrial function and metabolism would strengthen the manuscript's claims about immune cell function.

1. When analyzing the impact of tigecycline on certain T cell subsets, authors chose to focus on naïve and memory cells. However, they also claim that tetracyclines have an anti-inflammatory effect, which would likely target glycolytic highly proliferative effector cells?

2. It would be interesting to evaluate whether defects in proliferation solely rely on impaired T cell priming. This could be easily evaluated by adding tigecycline to culture after stimulation, but prior to first cell division.

3. Considering that OXPHOS is impaired in cells exposed to tigecycline, what is the effect on glycolysis? Do cells die because they are unable to meet their metabolic demands? Or would glycolysis rates increase as a compensatory mechanism?

4. The study would benefit from an alternative measurement of live mitochondrial translation in primary cells, which can be done using an adaptation of the Scenith technique with isolated mitochondria (Delaunay et al., 2022, Nature; Yazicioglu et al., 2023, Nature Immunology).

5. Authors mention that "The tigecycline binding site likely hinders the interaction of mitoribosome with the ribosomal recycling factor 1 (RRF1)", which might result in non-functional mitoribosomes prone to aggregation. Could authors address this point? Or at least evaluate whether mitochondrial mass/architecture is changed in tigecycline-treated cells.

6. Previous studies (e.g. Perry et al., 2021, Nature Metabolism) claim that tetracyclines promote cell survival and fitness in mitochondrial disease scenarios, also by inhibiting mitochondrial translation. It would be interesting if authors would discuss how treatment with different tetracyclines would result in divergent outcomes concerning cell viability.

Minor points: Western Blots should indicate molecular weight, flow cytometry plots need to have clear scale even in histograms, antibodies need to have their fluorophore described.

Version 1:

Reviewer comments:

Reviewer #1

(Remarks to the Author)

The authors have addressed most of my concerns.

Minor Comments:

Figure S3: The y-axis of the FSC curves requires labeling, including the 0.143 threshold.

Page 3, line 100: The published ribosome structures are not at atomic resolution. Can individual atoms be resolved in these structures?

Reviewer #3

(Remarks to the Author)

The authors have effectively addressed all comments raised by reviewers, enhancing the robustness of the study compared to its original version. They may consider incorporating some rebuttal figures as supplementary information - for example, rebuttal figure 3, which clearly demonstrates that mitochondrial mass remains unaffected and that the tigecycline phenotype depends on mitochondrial function.

REVIEWER COMMENTS

We would like to thank the reviewers for their time in evaluating our manuscript and raising pertinent questions, which, as described in our responses below, have been important in improving our manuscript.

Reviewer #1 (Remarks to the Author):

Shao et al. (2024) present a comparative study investigating the cytotoxic effects of various antibiotics on T cells, highlighting tigecycline, a third-generation tetracycline, as particularly potent. Their in vitro experiments showed that tigecycline at concentrations of 5 μ M inhibits mitochondrial translation without affecting cytosolic translation. Using cryo-electron microscopy (cryo-EM), they analyzed mitochondrial ribosomes from Jurkat T cells, revealing tigecycline's binding to the A-site of the small ribosomal subunit, preventing A-site tRNA binding, and identifying two additional binding sites in the large mitochondrial ribosomal subunit. These findings align with a recent study by Li et al. (Nat Commun, 2024), which also demonstrated tigecycline binding to both the small and large ribosomal subunits.

Major Points

The originality of Shao et al.'s study is questionable. Li et al.'s recent publication in Nature Communications has already demonstrated that tigecycline binds to both the small and large mitochondrial ribosomal subunits, inhibiting translation. This overlap in findings raises concerns about the additional value provided by Shao et al.'s work. Without further insights, it is unclear how this study will advance the field.

We appreciate the reviewer's concern regarding the alignment of our study with the recent publication by Li et al. (Nat Commun, 2024). However, we would like to emphasize that our study is an independent investigation conducted in parallel with Li et al.'s work, rather than following the previous study. While both studies address tigecycline's interaction with the mitoribosome, our findings provide distinct and complementary insights into this interaction, and highlight effects on the immune system. Our research demonstrates that primary T cells (especially upon activation) are particularly sensitive to tigecycline-induced mitochondrial translation inhibition due to their distinct metabolic profile and heavy dependence on mitochondrial function.

Employing different experimental systems for structural studies - HEK cells versus Jurkat cells - our group and Li et al. independently identified three binding sites for tigecycline on mitoribosomes at clinically achievable concentrations. The convergence of these findings, using mitoribosomes from different cell types and different experimental setups, provides robust validation of the structural insights.

This is particularly important as one of the reported sites is located on a large mitoribosome subunit (mtLSU site-1) in a region previously not known to be affected by antibiotic binding. Our analyses directly compare this newly identified binding site with its bacterial counterparts, revealing that the absence of methylations on the mitoribosome—unlike the modifications present on bacterial ribosomes—likely contributes to this unique binding. These analyses underscore key structural differences between mitochondria and bacteria not reported previously (page 12, lines 373-383).

Furthermore, our study provides the first structural characterization of mitoribosomes from T cells, offering unique insights into tissue/cell-specific aspects of mitoribosome structure and its cofactors (page 8, lines 259-262).

Together, our findings not only validate key structural insights through independent methods but also extend our understanding of tissue-specific effects of tigecycline, with implications both for antibiotic development and therapeutic applications.

Minor Points

The cryo-EM data processing workflow lacks sufficient detail (Fig S2). The authors should expand on this workflow for reproducibility and transparency. Specifically, including a representative micrograph and the major 2D classes used in their analysis would provide a better understanding of the data quality.

Thank you for this suggestion. We have updated the figure (Fig S3, old Fig S2) with more details of the workflow. A representative micrograph and the major 2D classes used are included.

The authors should enhance the labeling of the FSC curves. Additionally, they should also present map-model FSCs.

We regenerated and highlighted the label of the FSC curves as shown in Fig S3. We also added a figure of map-model FSC as Fig S4.

The manuscript does not specify the number of classes used in each 3D classification round. This information is important for evaluating the thoroughness and robustness of their structural analysis. Indeed, there is a discrepancy between the sum of particles with P site tRNA and empty P site when compared to the total number of particles used for the initial 3D refinement.

Thank you for pointing out this. We now included this information in the Method section, CryoEM data collection and analysis part:

Mask targeting the P-tRNA was generated to perform 3D variability analysis and subsequently classified into three particle clusters, with each representing different features of the monosome. As a result, a subset of 11,757 particles were excluded after this round due to a poor electron density map. Next, a total of 124,050 particles with clear density for the P-tRNA were subjected to a second round of 3D variability analysis for further classification into three particle clusters using the A-tRNA as a mask. Finally, after two steps of classification by 3D variability analysis, we could eventually obtain three major classes, class 1 with the majority of the particles ('Empty': 130,443 particles) lacked densities for the A-site and P-site tRNA on the monosome, class 2 ('P-tRNA only': 87,076 particles) lacked the occupancy for the A-tRNA and is represented by the monosome bound to P-tRNA only and mRNA, and class 3 ('A- and P-tRNA': 36,974 particles) contained monosome bound A-tRNA, P-tRNA, and mRNA (Fig. S3).

We also corrected the processed particles from 266,550 to 266,250 in the Method section and the Table 1.

In Figure 2B, the authors present an atomic model for mRNA spanning the entire mRNA channel. However, without access to the corresponding map and coordinates, it is impossible to assess whether the map's quality justifies building the mRNA model from the entry to the exit site. The authors should include a map-to-model fitting for mRNA and tRNA as a supplementary figure.

We generated a new figure showing the map-to-model fitting for mRNA and tRNA in our map as in Fig S5A.

How does the binding site in the SSU compare with that reported by Brodersen et al. 2000 (PMID: 11163189)?

In that paper, Brodersen et al. reported 2 binding sites of tetracycline in the SSU. The primary site is almost identical to the binding site of tigecycline in the mtSSU in our manuscript (Rebuttal Figure 1). This binding site is also conserved for other tetracyclines such as sarecycline as shown in Figure 3 in our manuscript. We did not observe the second binding site of tetracycline at the SSU site as reported by Brodersen et al. Since we discuss comparison of the mtSSU binding site with the binding of multiple antibiotics from the tetracycline family already in the text (in 'Tigecycline blocks aminoacyl-tRNA binding to the mtSSU A site') and present a direct comparison with the binding of tigecycline to E. coli 70S (Fig 3B), we decided not to include the comparison with Brodersen et al. in the manuscript. Instead, we added this reference in the sentence '*The interaction between tigecycline and the mtSSU is analogous to how tetracyclines bind to the small ribosomal subunits of various bacterial species (Fig. 3A-C)²⁸⁻³⁵.*' (page 10, 311-312).

Rebuttal Figure 1

Rebuttal Figure 1: The structural comparison between the tetracycline (orange) bound bacterial SSU (grey) (PDB: 1HNW) and tigecycline (purple) bound mtSSU (light yellow) (this study).

Reviewer #3 (Remarks to the Author):

Shao, Khawaja, Nguyen, and colleagues explore the impact of the antibiotic tigecycline on both immortalized and primary human T cells. Due to the endosymbiotic origins of mitochondria, it is well-established that mitochondrial ribosomes share several structural and functional similarities with bacterial ribosomes. Building on this understanding, as well as existing literature suggesting that antibiotics adversely affect mitoribosome function, the authors demonstrate that tigecycline compromises T cell survival and proliferation by impairing oxidative phosphorylation (OXPHOS). This is followed by a detailed structural characterization of the specific binding sites of tigecycline on mitoribosomes, employing single-particle cryo-electron microscopy. The results obtained explain the inhibitory effect of tigecycline on mitochondrial translation. While other recent studies have examined tigecycline's interaction with mitoribosomes, the significance of this work is preserved by its focus on immune cells. Further characterization of mitochondrial function and metabolism would strengthen the manuscript's claims about immune cell function.

We thank the reviewer for their positive comments and for recognizing the importance of studying this topic. While we combine structural analysis with basic immunological characterization, we acknowledge that further studies will be needed in the future to deepen our understanding.

1. When analyzing the impact of tigecycline on certain T cell subsets, authors chose to focus on naïve and memory cells. However, they also claim that tetracyclines have an anti-inflammatory effect, which would likely target glycolytic highly proliferative effector cells?

As shown in the updated figures (Figure 1I, S2E and S2F), our results demonstrate that treatment with 10 μ M tigecycline significantly inhibited proliferation across TCR-stimulated primary human CD4⁺ T cells, including naïve, central memory, and effector memory subsets. These results highlight that even highly proliferative effector cells generated by *in vitro* polyclonal stimulation depend upon upregulated OXPHOS, as well as glycolysis. Further *in vivo* and *ex vivo* studies are needed to determine how tetracyclines influence different immune cells and states across tissues and infectious contexts, although this is beyond the scope of this work.

2. It would be interesting to evaluate whether defects in proliferation solely rely on impaired T cell priming. This could be easily evaluated by adding tigecycline to culture after stimulation, but prior to first cell division.

Thank you for this suggestion. To address this, we expanded our proliferation experiments examining tigecycline's impact at 24 and 72 h post-activation. As shown in the updated figures (Figure S2A and S2B), tigecycline significantly reduced the proliferation of both CD4⁺ and CD8⁺ T cells when administered at either 24 or 72 h after stimulation. This supports that tigecycline's anti-proliferative effects extend beyond initial T cell priming.

3. Considering that OXPHOS is impaired in cells exposed to tigecycline, what is the effect on glycolysis? Do cells die because they are unable to meet their metabolic demands? Or would glycolysis rates increase as a compensatory mechanism?

In our experiments, adequate glycolysis rates (added to the manuscript as **Figure S1F**) trended towards being decreased after 10 uM treatment, most likely due to cell death in the culture. However, 5 uM treatment trended towards an increase in glycolysis compared to untreated, stimulated controls, supporting that compensatory increases in glycolysis can occur at less toxic concentrations. However, given the strength of OXPHOS inhibition at these concentrations, this is not sufficient to promote survival of proliferating T cells.

4. The study would benefit from an alternative measurement of live mitochondrial translation in primary cells, which can be done using an adaptation of the Scenith technique with isolated mitochondria (Delaunay et al., 2022, Nature; Yazicioglu et al., 2023, Nature Immunology).

Thank you for suggesting an analysis of live mitochondrial translation in primary cells. For this, we employed mito-FUNCAT-FACS, a combination of click chemistry and flow cytometry, to assess mitochondrial translation at the single cell level (as in Kimura et al, *RNA* 2022). We decided to use this method over Scenith as we have optimised it in our group and it does not require isolation of mitochondria. This approach allowed us to specifically monitor newly synthesized mitochondrial peptides in distinct T cell subpopulations within TCR-stimulated PBMCs. As shown in the updated figures (**Figure S1C and S1D**), the results demonstrated that treatment with tigecycline led to decreased mitochondrial translation in both CD4⁺ and CD8⁺ T cells, as evidenced by reduced fluorescent labeling of nascent mitochondrial peptides.

Of note - to investigate whether this method quantitatively correlates with S35 measurements, we conducted side-by-side measurements of HEK cell translation under tigecycline treatment (same concentration and time of incubation) using both approaches. While both methods demonstrated a significant reduction in translation, the results cannot be directly compared quantitatively (Rebuttal Figure 2). Specifically, we observed that the decrease in translation detected by FUNCAT (A) was smaller than that visualized by S35 labeling (B). This discrepancy arises from differences in normalization: S35 measurements are normalized to the background signal, whereas FUNCAT is normalized to the signal from a control (treatment with a high concentration of anisomycin).

Rebuttal Figure 2

5. Authors mention that “The tigecycline binding site likely hinders the interaction of mitoribosome with the ribosomal recycling factor 1 (RRF1)”, which might result in non-functional mitoribosomes prone to aggregation. Could authors address this point? Or at least evaluate whether mitochondrial mass/architecture is changed in tigecycline-treated cells.

Thank you for raising this important point. We are currently conducting comprehensive analyses to address the mechanistic consequences of tigecycline binding to mitoribosome .

Our investigation using MitoTracker Green staining revealed no significant changes in mitochondrial mass of PBMCs following 10 uM tigecycline treatment, both at 24-hour and 7-day timepoints (Rebuttal Figure 3). This stability of mitochondrial content persists despite inhibition of mitochondrial protein synthesis.

To specifically examine potential mitoribosome aggregation, we performed sucrose density gradient analysis followed by Western blot detection of mitoribosomal proteins. After 1 hour exposure to tigecycline, we observed a slight increase in the intensity of the monosome peak, but likely no formation of higher-order complexes (Rebuttal Figure 4).

In summary, while these analyses may provide insights into tigecycline's effects on mitochondrial parameters, we acknowledge that additional studies are needed to fully characterize the effect of tigecycline and the RRF1 mechanism. We are currently investigating potential activation of mitoribosome quality control pathways after antibiotic treatment. This is beyond the scope of this study, and we do not believe adding these preliminary data would strengthen the manuscript. Instead, we decided to moderate the sentence:

‘The tigecycline binding site likely hinders the interaction of mitoribosome with the ribosomal recycling factor 1 (RRF1) , which might result in non-functional mitoribosomes prone to aggregation’ → ‘The tigecycline binding site likely hinders interactions between mitoribosomes and ribosomal recycling factor 1 (RRF1), which may result in non-functional mitoribosomes’.

Rebuttal Figure 3

Rebuttal Figure 4

6. Previous studies (e.g. Perry et al., 2021, Nature Metabolism) claim that tetracyclines promote cell survival and fitness in mitochondrial disease scenarios, also by inhibiting mitochondrial translation. It would be interesting if authors would discuss how treatment with different tetracyclines would result in divergent outcomes concerning cell viability.

Thank you for raising this interesting question.

One of critical distinction lies in the cellular models investigated. T lymphocytes possess unique bioenergetic requirements and rely heavily on mitochondrial function for their activation and expansion. In contrast, the respiratory chain mutant fibroblasts studied by Perry et al. (MELAS, ND1, and Rieske KO) may have metabolic and compensatory mechanisms that allow them to benefit from partial translation inhibition. We report a stronger effect of doxycycline in Jurkat cells than HeLa cells, highlighting cell-specific differences (Fig S1). Future single-cell and spatial sequencing studies of tissue lesions during infection and antibiotic treatment is one approach to the question from an immunological perspective.

Perry et al. demonstrated that tetracyclines exhibit a hormetic dose-response curve, where low concentrations (around 1 µM) partially inhibit mitochondrial translation in patient fibroblasts, triggering adaptive responses that enhance cellular fitness. In contrast, our study examined slightly higher therapeutic concentrations (2 - 10 µM) of tigecycline, a seemingly more potent tetracycline derivative than doxycycline and tetracycline investigated in Perry et al. This resulted in stronger inhibition of mitochondrial translation in T cells potentially overwhelming any adaptive mechanisms.

It is very interesting that the in vivo model chosen by Perry et al characterizes CNS lesions driven by immune cells. Simon Johnson and colleagues have shown that pharmacologic agents targeting the immune system can prevent disease in the *Ndufs4* KO model of Leigh syndrome, indicating that the immune system plays a causal role in the pathogenesis of at least this form of mitochondrial disease (Stokes et al, JCI Insights, 2022).

It is tempting to speculate that treatment with doxycycline in Perry et al, which as the authors stated, corrects neuroimmune and inflammatory proteins, stems from direct inhibition of the immune response due to the inhibition of mitochondrial translation, as in our model.

These findings collectively suggest that the therapeutic window for tetracycline treatment may be narrower than previously appreciated and highly dependent on the pathological context. The dose-dependent transition from beneficial to detrimental effects in certain cells emphasizes the importance of careful dosing strategies and consideration of tissue-specific responses in clinical applications.

Rather than contradicting each other, these studies enhance our understanding of the complex relationship between tetracycline treatment and cellular homeostasis in the context of mitochondrial disease and immune activation.

We added a paragraph to the discussion referencing Perry et al.

'Our data encourage the investigation of anti-tumor immunity in patients receiving such treatments but also illustrate different cell types respond differently to OXPHOS inhibition, depending on the genetic program. Indeed, low doses of tetracyclines promoted the survival of fibroblasts derived from mitochondrial disease patients, which may have different compensatory mechanisms to cope with reductions in OXPHOS²⁴. Future studies should, therefore, investigate how antibiotics affect different cells in vivo across clinical contexts.'

Minor points: Western Blots should indicate molecular weight, flow cytometry plots need to have clear scale even in histograms, antibodies need to have their fluorophore described.

Thank you, we have updated the related figures accordingly.